

# Integrating microRNA expression, miRNA-mRNA regulation network and signal pathway: a novel strategy for lung cancer biomarker discovery

Renqing Nie, Wenling Niu, Tang Tang, Jin Zhang and Xiaoyi Zhang

Faculty of Environment and Life, Beijing University of Technology, Beijing, China

Corresponding author
Xiaoyi Zhang,
zhangxiaoyi@bjut.edu.cn

## ABSTRACT

**Background:** Since there are inextricably connections among molecules in the biological networks, it would be a more efficient and accurate research strategy to screen microRNA (miRNA) markers combining with miRNA-mRNA regulatory networks. The independent regulation mode is more "fragile" and "influential" than the co-regulation mode. miRNAs can be used as biomarkers if they can independently regulate hub genes with important roles in the PPI network, simultaneously the expression products of the regulated hub genes play important roles in the signaling pathways of related tissue diseases.

**Methods:** We collected miRNA expression of non-small cell lung cancer (NSCLC) from The Cancer Genome Atlas (TCGA) database and the Gene Expression Omnibus (GEO) database. Volcano plot and signal-to-noise ratio (SNR) methods were used to obtain significant differentially expressed (SDE) miRNAs from the TCGA database and GEO database, respectively. A human miRNA-mRNA regulatory network was constructed and the number of genes uniquely targeted (NOG) by a certain miRNA was calculated. The area under the curve (AUC) values were used to screen for clinical sensitivity and specificity. The candidate markers were obtained using the criteria of the top five maximum AUC values and NOG ≥ 3. The protein–protein interaction (PPI) network was constructed and independently regulated hub genes were obtained. Gene Ontology (GO) analysis and KEGG pathway analysis were used to identify genes involved in cancer-related pathways. Finally, the miRNA which can independently regulate a hub gene and the hub gene can participate in an important cancer-related pathway was considered as a biomarker. The AUC values and gene expression profile analysis from two external GEO datasets as well as literature validation were used to verify the screening capability and reliability of this marker.

**Results:** Fifteen SDE miRNAs in lung cancer were obtained from the intersection of volcano plot and SNR based on the GEO database and the TCGA database. Five miRNAs with the top five maximum AUC values and NOG ≥ 3 were screened out. A total of 61 hub genes were obtained from the PPI network. It was found that the hub gene *GTF2F2* was independently regulated by *miR-708-5p*. Further pathway analysis indicated that *GTF2F2* participates in protein expression by binding with polymerase II, and it can regulate transcription and accelerate tumor growth. Hence, *miR-708-5p* could be used as a biomarker. The good screening capability and reliability of *miR-708-5p* as a lung cancer marker were confirmed by AUC values and

gene expression profiling of external datasets, and experimental literature. The potential mechanism of *miR-708-5p* was proposed.

**Conclusions:** This study proposes a new idea for lung cancer marker screening by integrating microRNA expression, regulation network and signal pathway. *miR-708-5p* was identified as a biomarker using this novel strategy. This study may provide some help for cancer marker screening.

## INTRODUCTION

Lung cancer is the leading cause of cancer-related deaths worldwide (*Siegel, Miller & Jemal, 2017*), and non-small cell lung cancer (NSCLC) accounts for the majority of new diagnoses. NSCLC is inherently incurable although many strategies have been proposed to improve patient survival (*Fassina, Cappellesso & Fassan, 2011*). Most patients are diagnosed with advanced disease, half of them with distant metastases at the time of initial diagnosis (*Siegel, Miller & Jemal, 2019*), so patients often miss the best time for surgical and other treatments, resulting in a poor prognosis with a 5-year survival rate of only 16% to 18% (*Tanoue et al., 2015*). However, if patients can be diagnosed early and receive treatment promptly, the 5-year survival rate can be increased to 45% to 65%(*Ettinger et al., 2010*). Therefore, it has become increasingly important to detect lung cancer earlier.

The early diagnosis of lung cancer mainly relies on imaging, cytology and biochemical examination, but due to the limitation of the current treatment level, the false positive rate is very high (*Xi et al., 2019*). The gold standard of pathological examination is fiberoptic bronchoscopy and percutaneous lung biopsy (*Diederich, 2009*), but these examinations have the disadvantages of invasiveness and poor compliance. Tumor marker tests, such as carcinoembryonic antigen (CEA), cytokeratin 19 fragment, squamous epithelial cell carcinoma (SCC) antigen, carcinoma antigen 125, neuron-specific enolase (NSE), have been widely used in the diagnosis of lung cancer; however, the sensitivity and specificity of these markers are not high (*I & Cho, 2015*). With the rapid development of high-throughput technologies, more and more biological data have been shown to be applicable to the prediction of cancer markers. Many studies have shown that miRNAs are closely correlated with the development of diseases, especially cancer (*Elliot et al., 2019*; *Sun et al., 2018*). The expression levels of miRNAs are statistically significantly different in the sera of patients with NSCLC compared to normal individuals (*Liang et al., 2020*; *Ying et al., 2020*), and nearly half of the annotated human miRNAs are located at vulnerable and critical points in the genome (*Calin et al., 2004*). miRNAs can exist stably without degradation in human tissues, blood and body fluids (*Benz et al., 2016*). They are small molecules and are specific in tissue expression and temporal expression, and can easily and accurately reflect the evolutionary pattern of disease development (*Abrahamsson & Dabrosin, 2015*), etc. Thus, miRNAs are promising markers for tumor diagnosis in clinical practice.

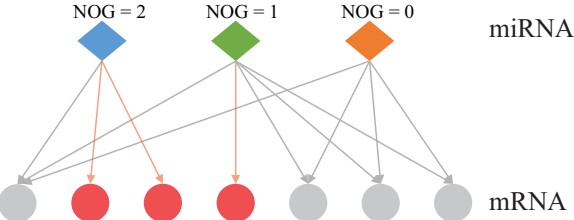

**Figure 1 Two regulatory types of miRNAs in the human miRNA-mRNA network.** The red circles represent mRNAs regulated by a single miRNA, the grey circles represent mRNAs regulated by multiple miRNAs. NOG, the number of genes uniquely targeted.

At present, most of the methods for identifying miRNA markers only consider the changes in the expression level of miRNA itself, and do not involve its regulatory genes. An individual biomolecule can't analyze the underlying mechanisms of various biological phenomena, but well-understood biological networks could lead to revealing the inherent laws of life activities at the systemic level (*Menche et al., 2015*). The differences in miRNA expression levels can be reflected in the expression levels of the target genes regulated by miRNA (*Bam et al., 2018*), therefore, the computational biology research method combined with miRNA-mRNA regulatory networks would be a more efficient and accurate research strategy, and it can also provide new ideas for finding cancer markers. Co-regulatory effects of miRNA-mRNA regulatory networks are considered in some studies, but the results are somewhat influenced by the training set data (*Zhang & Shen, 2013*). *Zhang et al. (2014)* proposed the independent regulation ability of miRNA. Figure 1 shows the two types of regulation modes of miRNA-mRNA: independent regulation mode means an mRNA is regulated by a unique miRNA, co-regulation mode means an mRNA can be regulated by multiple miRNAs. The miRNA which can independently regulate multiple mRNAs has a larger number of genes uniquely targeted (NOG) value. *Lin et al. (2018a)* reported that miRNAs with significant NOG in miRNA-mRNA networks have stronger independent regulatory abilities, independent regulatory ability is fragile and important for the stability of biological networks. The statistical evidence suggests that most miRNAs that can be used as biomarkers have significantly larger NOG values, namely, the independent regulatory ability would be an important network feature of miRNA biomarkers (*Lin et al., 2018b*; *Lin, Yuan & Shen, 2016*).

In addition, we believe that the regulatory ability of a miRNA also lies in the importance of the genes regulated by the miRNA. The importance of the gene can be divided into two aspects: one is whether the expression product of the gene is a core node in the protein-protein interaction (PPI) network, highly connected hub gene plays an important role in the biological processes; and the other is whether the expression product of the gene plays a role in an important signaling pathway related to the disease.

Considering that the independent regulation mode of miRNA-mRNA is more "fragile", the abnormal expression of the only miRNA will lead to a large change of this "fragile" structure, if the expressed protein of the regulated gene act as a key node in the PPI network, the dysregulation of the sole miRNA-gene interaction would result in a dramatic impact on intracellular function, if this gene plays an essential role in a disease-related

signaling pathway, the disorder of independent regulation would inevitably lead to disease onset and progression. This shows the huge influence of this independent regulation mode. Therefore, a new idea for miRNA marker screening is proposed here: a hub gene can be independently regulated by a significant differentially expressed (SDE) miRNA, and the express product of the hub gene plays an important role in the signaling pathway of the relevant tissue disease, then the miRNA can be used as a biomarker.

This study is going to use computational recognition to determine the SDE miRNAs between lung cancer and paraneoplastic tissues. A regulatory network will be constructed and NOG value will be obtained. Then, the sensitivity and specificity of those SDE miRNAs would be evaluated by area under the curve (AUC) value. The miRNAs with the top five maximum AUC values and NOG ≥ 3 would be picked out as candidate markers. Then, the importance of the genes regulated by candidate markers was further investigated. Firstly, a PPI network will be constructed using the target genes of these five candidate miRNAs, and the hub genes independently regulated by candidate miRNA in the PPI network will be found out, and then lung cancer-related signaling pathways participated by these hub genes would be analyzed. If there is a hub gene that is independently regulated by a miRNA, and the expression product of this gene plays an important role in lung cancer-related signaling pathways, then the miRNA can be used as a biomarker. Finally, AUC values and gene expression profile analysis based on two external GEO datasets as well as literature validation were used to verify the screening ability and rationality of the marker. The potential mechanism of the biomarker will be assessed. The workflow is displayed in Fig. 2.

## MATERIALS & METHODS

### Data download and preprocessing

In order to eliminate the background difference of the obtained sample data, two different databases were selected. miRNA-seq isoform quantification data was obtained from The Cancer Genome Atlas (TCGA, https://portal.gdc.cancer.gov/) and GSE102286 dataset of the Gene Expression Omnibus (GEO, https://www.ncbi.nlm.nih.gov/geo/). The data from cancer tissues and paraneoplastic tissue were paired up. Two datasets (GSE56036 and GSE36681) were downloaded from the GEO as validation sets. The GSE56036 dataset contains 29 NSCLC samples and 19 normal samples. We selected all 56 pairs of fresh frozen samples of NSCLC tissue and normal tissue from the GSE36681 dataset.

The data was pre-processed. Firstly, the miRNAs with missing values of more than 20% in all samples were removed. In additional, the Z-score method was used to find outliers and replace the rest of missing values with medians. Due to the large gap between the number of cancer samples and paraneoplastic samples, so smote algorithm was used to balance the data. Finally, the max-min normalization method was used to normalize the dataset.

### Obtaining candidate biomarkers
#### *Computational identification of SDE miRNAs*

As a special scatter map, volcano plot was used to quickly identify those individuals who have differences in the mass data, and it can intuitively display the data changes. There are
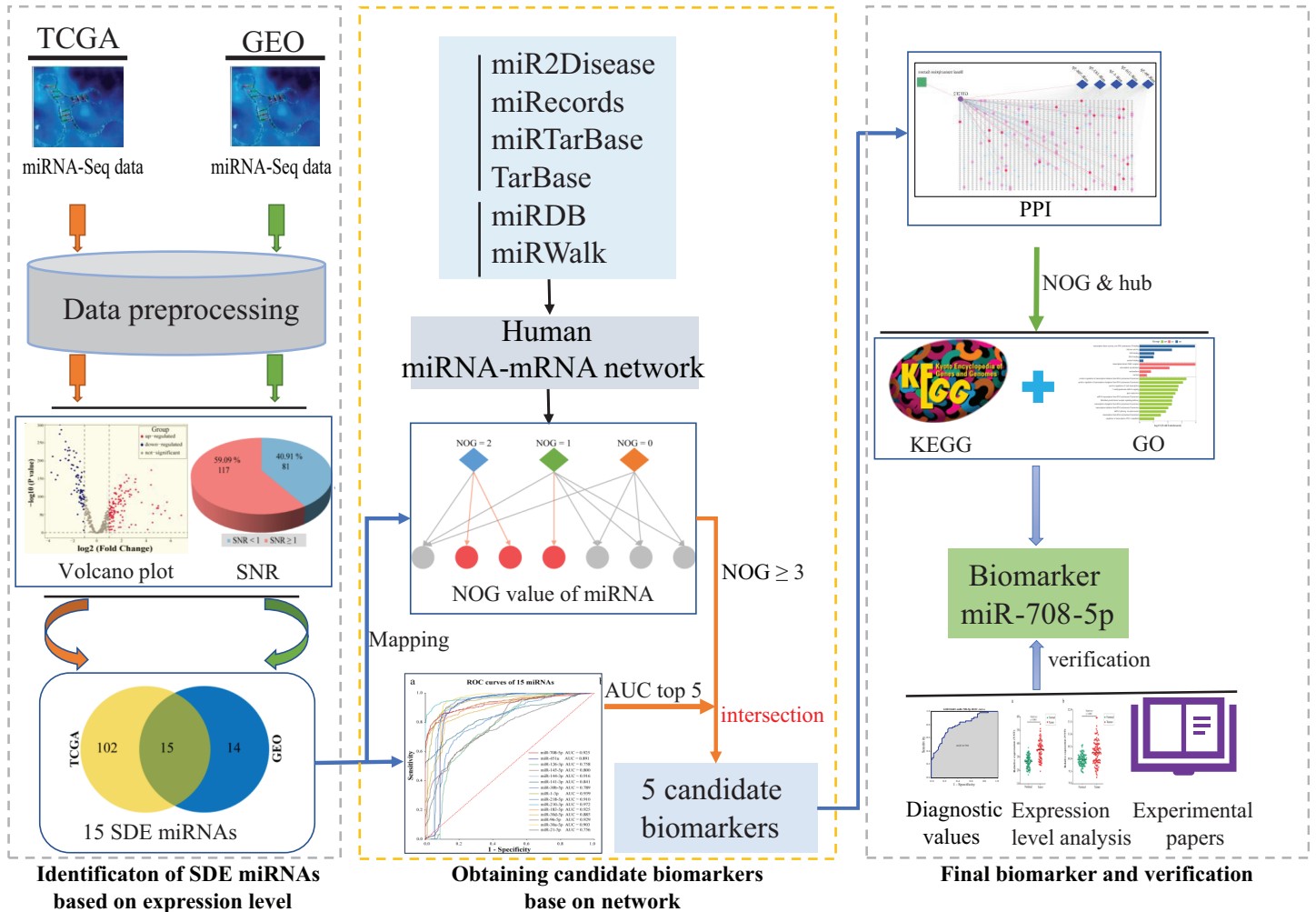

**Figure 2 The workflow of this study.** SDE, significant differentially expressed; NOG, the number of genes uniquely targeted.

two important indexes of volcano plot, which are fold change (FC) and *p*-value (*p*-value). Threshold criteria of screening statistics: |log2 (FC) | ≥ 1, *p*-value < 0.05. The results of the Volcano plot were used for further screening by the signal-to-noise ratio (SNR) test (Eq. 1). The SNR method is an effective method to remove noise, evaluate the classification ability of genes according to the scoring level, discard the lower ranking genes and keep only the subsets with high scores. It is a coarse-grained feature selection method.

$$SNR = \frac{u_+(i) - u_-(i)}{\sigma_+(i) - \sigma_-(i)} \tag{1}$$

In the above formula, $u_+(i)$ and $u_-(i)$ represent the average expression value of gene i in normal and tumor tissue samples, respectively. $\sigma_+(i)$ and $\sigma_-(i)$ represent the standard deviation of gene i in normal tissue samples and tumor tissue samples, respectively.

### Construction of human miRNA-mRNA regulatory network and calculation of NOG values

To obtain the independent regulatory capabilities (NOG value) of miRNA candidate markers, six commonly used public databases were used to construct human miRNA-mRNA regulatory networks. Among them, the data in the four databases of miR2Disease (http://www.mir2disease.org/7), miRecords (http://mirecords.biolead.org/), miRTarBase (https://mirtarbase.cuhk.edu.cn/~miRTarBase/miRTarBase_2022/php/index.php), and TarBase (http://www.microrna.gr/tarbase) mainly come from biological experiments such as high-throughput and low-throughput. The data of the other two databases, miRDB (http://mirdb.org) and miRWalk (http://mirwalk.umm.uni-heidelberg.de), mainly come from computer algorithm prediction. The miRNA-mRNA regulatory network can be regarded as a unidirectional regulatory network. Some genes in this regulatory network are regulated only by a single miRNA, which means this miRNA has independent regulatory capabilities for this gene. If a miRNA regulates several genes independently, the number will be used as the NOG value of this miRNA. Generally, the NOG value of this miRNA is larger, the ability of the miRNA to independently regulate genes is stronger. In this study, the value of NOG $\geq$ 3 indicates that the miRNA has strong independent regulatory capabilities, this criterion was used to screen candidate markers.

### Sensitivity and specificity analysis using ROC

For SDE miRNAs, the ROC curve was plotted by the GraphPad Prism 8 software (http://www.graphpad.com) to identify the miRNAs which have good clinical sensitivity and specificity. The AUC can be used as a quantification standard for diagnostic classification capabilities. The larger the AUC value, the higher the diagnostic classification value. Criteria for screening diagnostic miRNA candidate markers: AUC $\geq$ 0.9. Top 5 miRNAs were selected for follow-up research.

## Obtaining biomarkers

### Construction of PPI network

Target genes of candidate miRNAs were obtained using the constructed human miRNA-mRNA network. Interactions between target gene products were constructed using the Search Tool for the Retrieval of Interacting Genes/Proteins (STRING; version 11.0) database and PPI networks were constructed using Cytoscape software (version 3.8.0; https://cytoscape.org/). The combined score (magnitude of the probability of protein interaction) should be greater than 0.9 (highest confidence). Nine methods of the CytoHubba—maximal clique centrality (MCC), the density of maximum neighborhood component (DMNC), maximum neighborhood component (MNC), degree, edge percolated component (EPC), closeness, radiality, betweenness and stress were used to calculate the score, and the upper quartiles of the scores were calculated separately. Genes with scores greater than the upper quartile for all nine methods were selected as hub genes.

### Bioinformatics analysis

To gain insight into whether the expression products of those hub genes play a role in important disease-related signaling pathways, Gene Ontology (GO) analysis was

performed using the DAVID (https://david.ncifcrf.gov/), and the online enrichment tool of KEGG (http://www.genome.jp/kegg/tool/map_pathway2.html) was used to perform the pathway analysis.

### Finding biomarkers

A hub gene which is independently regulated by a miRNA, and the expression product of this gene plays an important role in lung cancer-related signaling pathways, then the miRNA can be used as a biomarker.

## Validation of the screening ability and reasonableness of biomarker

### Reliability validated using expression profiling

The differences in miRNA expression levels can be reflected in the expression levels of the target genes regulated by miRNA. So, the expression changes in lung cancer tissues of the marker miRNA and the hub gene independently regulated by it were verified. Expression data were downloaded from the TCGA database, and paired t-tests were performed on paired lung cancer and paraneoplastic samples using GraphPad Prism 8. Using Pearson correlation, we explored the association between miR-708-5p and *GTF2F2* expression and plotted the correlation with R package ggstatsplot (https://github.com/IndrajeetPatil/ggstatsplot).

### Evaluating the performance of the marker miRNA using external datasets

The GSE56036 and GSE36681 datasets were obtained from GEO database for the expression level evaluation of miR-708-5p. Then ROC analyses were performed using GraphPad Prism 8 to evaluate the diagnostic value of miR-708-5p.

### Reliability validated by literatures

The biomarkers were submitted to PubMed database. Literature screening criteria for a certain miRNA: "miR-XXX"[All Fields] AND "humans"[MeSH Terms]. Literature screening criteria for whether miRNA is associated with cancer: "Neoplasms"[Mesh] AND "miR-XXX"[All Fields] AND "humans"[MeSH Terms]. Literature screening criteria for the association of amiRNA with lung cancer: "Lung Neoplasms"[Mesh] AND "miR-XXX"[All Fields] AND "humans"[MeSH Terms].

## RESULTS

### Preprocessing results

A total of 993 lung cancer tissue samples and 91 paraneoplastic tissue samples were downloaded from the TCGA database, each sample contains 2,275 mature miRNAs. A total of 59 lung cancer tissue samples and 59 paraneoplastic tissue samples were downloaded from the GEO database (GSE102286), each sample contains 654 mature miRNAs.

After sample balancing and three-step data preprocessing, there were 993 lung cancer tissue samples and 993 paraneoplastic tissue samples in the TCGA dataset, and the
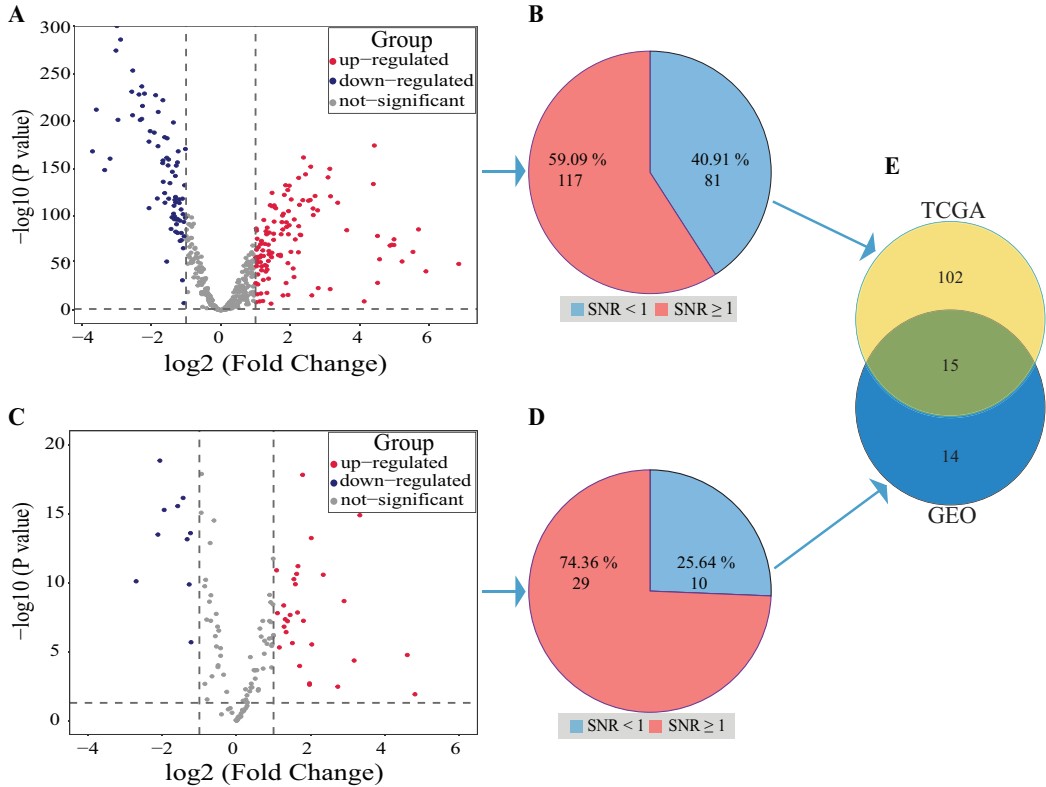

**Figure 3  Results of SDE miRNAs.** Volcano (A) and signal-to-noise ratio (B) from TCGA, volcano (C) and signal-to-noise ratio (D) from GEO, A and C, significantly down-regulated and up-regulated miRNAs are painted in blue and red. Venn diagram (E) showed the overlapping results of SDE miRNAs from TCGA and GEO.               

miRNAs in each tissue sample were reduced to 520. There were 59 lung cancer tissue samples and 59 paraneoplastic tissue samples, and the miRNAs in each sample were reduced to 125 in the GSE102286 dataset.

## Obtaining candidate biomarkers
### Computational identification of SDE miRNAs

Volcano plot result showed that 122 miRNAs in the TCGA database were up-regulated and 76 miRNAs were down-regulated, a total of 198 miRNAs with differential expression (Fig. 3A). The SNR result showed that 117 miRNAs with differential expression were screened from the 198 miRNAs (Fig. 3B; Table S1).

In the dataset GSE102286, the volcano plot results showed 29 miRNAs up-regulated and 10 down-regulated, totaling 39 miRNAs with differential expression (Fig. 3C). The SNR results showed that 29 miRNAs with differential expression were screened from the 39 miRNAs (Fig. 3D; Table S2).

By using Venn diagrams, the outputs of the two databases were intersected as a set of miRNAs with significant differences. As shown in Fig. 3E, 15 SDE miRNAs (six up-regulated and nine down-regulated) were identified totally (Table 1).

**Table 1 Computational identification of 15 SDE miRNAs.**

| miRNA | P-value | log2 (FC) | SNR | Regulation |
|-------|---------|-----------|-----|------------|
| hsa-miR-21-5p | 3.62E−87 | 1.019878 | 1.991775 | Up |
| hsa-miR-30a-5p | 2.91E−202 | −2.95875 | 1.503804 | Down |
| hsa-miR-96-5p | 1.13E−152 | 2.589762 | 1.151469 | Up |
| hsa-miR-30d-5p | 2.28E−174 | −1.8064 | 2.275344 | Down |
| hsa-miR-183-5p | 1.42E−144 | 2.43937 | 1.126407 | Up |
| hsa-miR-210-3p | 4.49E−175 | 4.419388 | 1.188131 | Up |
| hsa-miR-218-5p | 5.53E−217 | −2.25307 | 1.9371 | Down |
| hsa-miR-1-3p | 1.94E−254 | −2.53346 | 3.092538 | Down |
| hsa-miR-30b-5p | 6.69E−103 | −1.35847 | 1.673761 | Down |
| hsa-miR-141-3p | 5.45E−125 | 1.475862 | 1.273509 | Up |
| hsa-miR-144-3p | 3.63E−161 | −3.18641 | 1.282499 | Down |
| hsa-miR-145-5p | 2.16E−111 | −1.34576 | 1.850366 | Down |
| hsa-miR-126-3p | 9.74E−87 | −1.43099 | 1.124842 | Down |
| hsa-miR-451a | 4.11E−149 | −3.34021 | 1.172588 | Down |
| hsa-miR-708-5p | 2.22E−141 | 3.087151 | 1.051665 | Up |

**Note:**
FC, Fold change; SNR, signal-to-noise.

### Results of human miRNA-mRNA regulatory network construction

The human miRNA-mRNA regulatory network involves 1,242 miRNAs, 14,995 target genes, and 325,240 action pairs. As shown in Fig. 4A, 554 (44.6%) miRNAs in the regulatory network have independent regulatory capabilities, and 232 (18.7%) miRNAs have strong independent regulatory capabilities (NOG ≥ 3). In Fig. 4B, the distribution of the NOG values for miRNAs in the human miRNA-mRNA network followed a power-law distribution.

There are 10 miRNAs (miR-183-5p, miR-21-5p, miR-210-3p, miR-708-5p, miR-145-5p, miR-126-3p, miR-30a-5p, miR-218-5p, miR-96-5p and miR-1-3p) with NOG ≥ 3 in 15 SDE miRNAs (Fig. 4C).

### Sensitivity and specificity analysis using ROC

To further evaluate the clinical diagnostic sensitivity and specificity of 15 SDE miRNAs, the ROC curve was used. There are eight miRNAs with AUC ≥ 0.9, they are miR-210-3p, miR-1-3p, miR-96-5p, miR-183-5p, miR-708-5p, miR-144-3p, miR-218-5p and miR-30a-5p (Fig. 4D).

### Candidate biomarkers obtained from the intersection of NOD and ROC

SDE miRNAs with top five AUC ranking (AUC ≥ 0.9) and NOD ≥ 3 were considered as miRNA candidate markers for lung cancer. Five candidate markers, namely, miR-210-3p, miR-1-3p, miR-96-5p, miR-183-5p and miR-708-5p, were obtained. They have good diagnostic effects and also have strong independent regulation abilities.

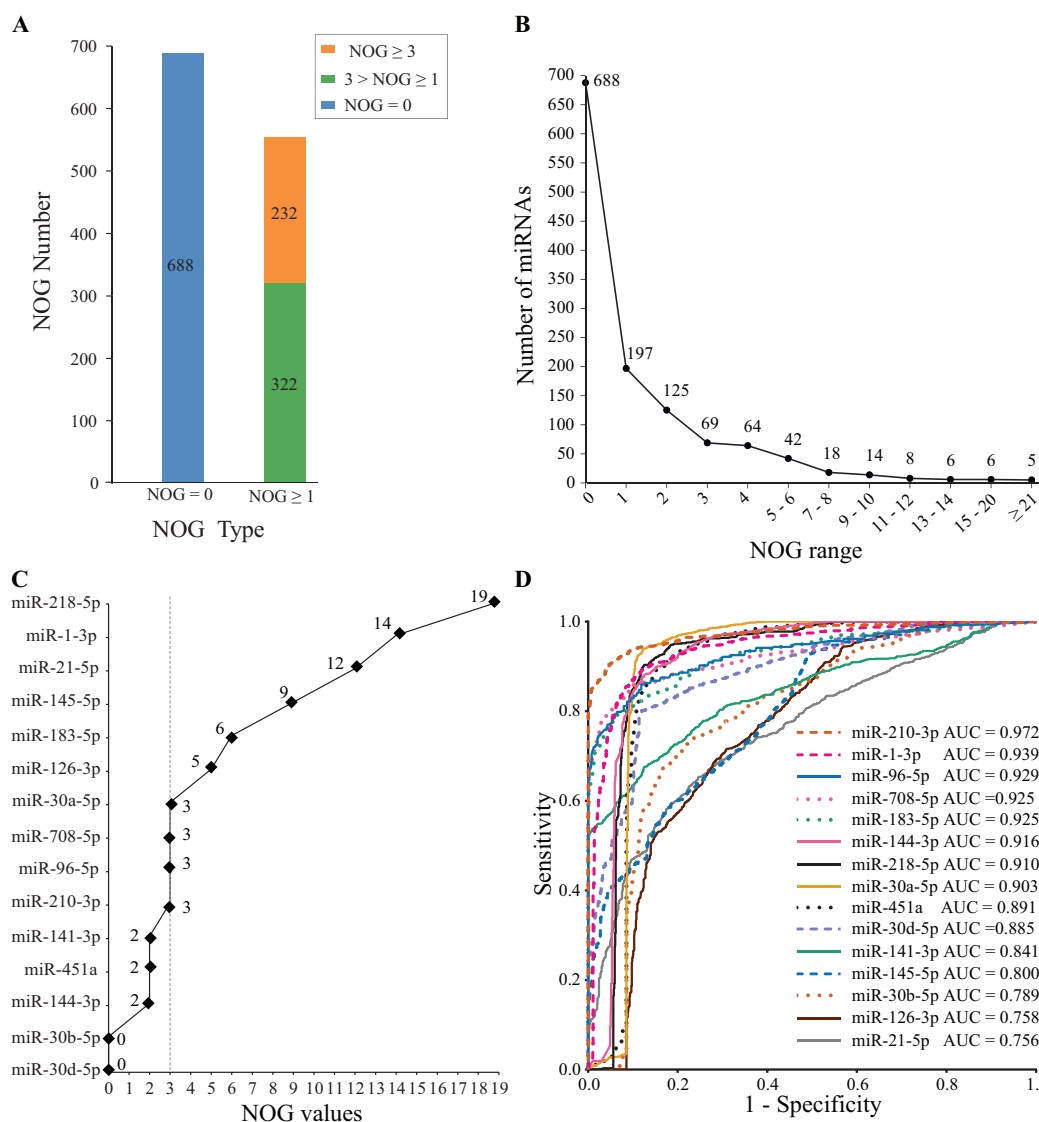

**Figure 4 NOG statistics of miRNAs in human miRNA-mRNA network and ROC curves of SDE miRNAs.** (A and B) The NOG distribution of miRNAs in the human miRNA-mRNA regulatory network. (C) NOG values of 15 SDE miRNAs. (D) ROC curves for 15 SDE miRNAs were plotted based on the TCGA data.

## Obtaining biomarkers

### PPI network construction and hub gene analysis

The PPI network was constructed by selecting all the target genes of five miRNAs with a combined score > 0.9 based on the STRING database. A node in the PPI network represents a protein, and the protein–protein interactions are presented by the link. The result is showed in Fig. 5 (Table S3).

The PPI network consisted of 1,807 nodes and 15,634 edges. A total of 61 hub genes were screened out (Table S4). Among these hub genes, *GTF2F2* (purple circle in Fig. 5),

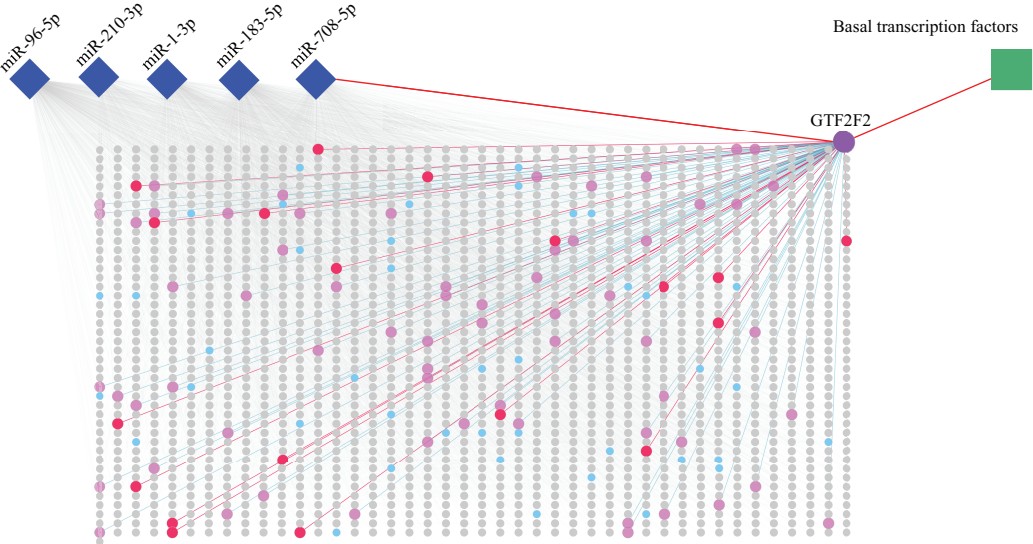

**Figure 5 PPI of five miRNA target genes.** The diamonds represent miRNAs, the circles stand for genes, and the square stands for a pathway. The rectangular area in the diagram is the PPI network. Purple circle, red circles and blue circles are hub genes. The purple circle (*GTF2F2*) is both hub gene and gene independently regulated by *miR-708-5P*. The red circles represent genes that are both hub genes and interact with *GTF2F2*. The pink circles represent genes that interact with *GTF2F2* but are not hub genes.                                           

independently regulated by miR-708-5p, has a high degree of connectivity, its degree is 83 (red circles and pink circles).

### Bioinformatics analysis

To investigate whether the gene *GTF2F2* is involved in lung cancer-related pathways, functional analysis of *GTF2F2* was performed using DAVID and the results was shown in Fig. 6. The biological processes (BP) in which *GTF2F2* was involved were: transcriptional elongation from the RNA polymerase II promoter, regulation of transcription, DNA-templated, positive regulation of viral transcription. The main molecular functions (MF) were: transcription factor activity, core RNA polymerase II binding, ATP binding, protein binding. The cellular components (CC) were: nucleoplasm, nucleus, transcription factor TF2F complex.

KEGG (Kyoto Encyclopedia of Genes and Genomes) pathway of *GTF2F2* was analyzed, and the results showed that the target gene *GTF2F2* was involved in the pathway of hsa03022: Basal transcription factors. *GTF2F2* binds with RNA polymerase II, and then participates in protein expression, which could regulate transcription and accelerate tumor growth (*Kalkat et al., 2018*).

### The final biomarkers

In a word, we found that the target gene *GTF2F2*, which was independently regulated by miR-708-5p, also existed as a key node in the PPI network, and it is in the pathway associated with cancer. According to our new idea for miRNA marker screening, miR-708-5p could be considered as a biomarker.

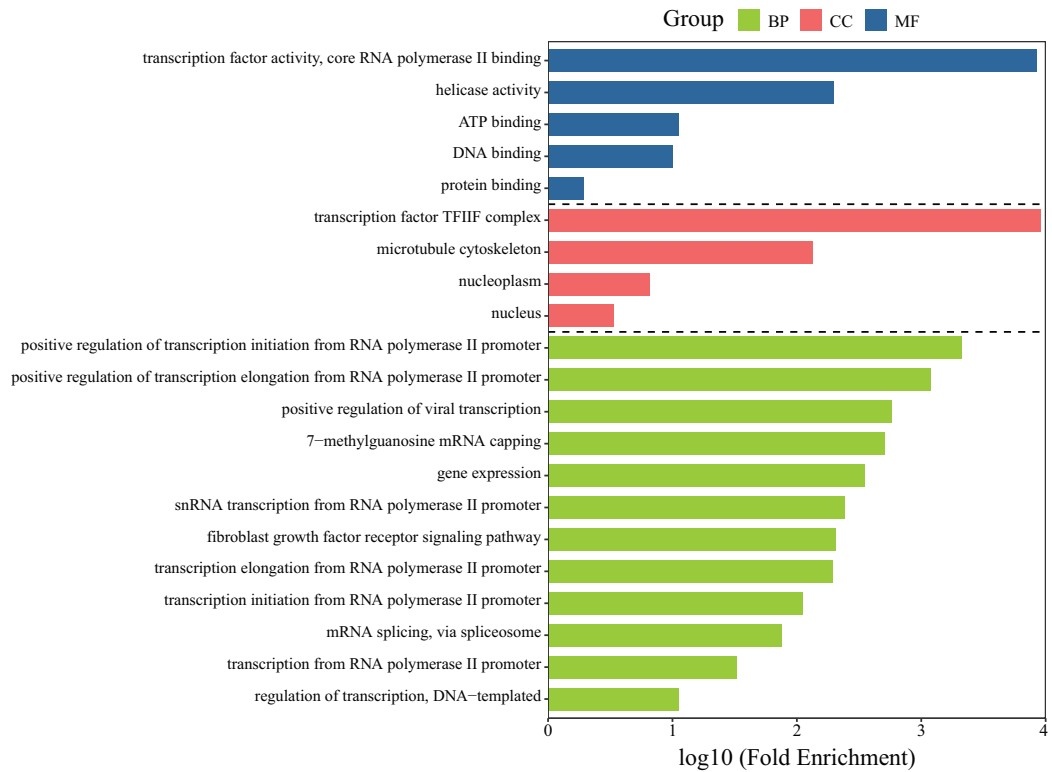

**Figure 6 GO functional analysis of *GTF2F2*.** BP, Biological Process; MF, Molecular Function; CC, Cellular Component.

## Validation of the screening ability and reliability of the biomarker
### *Reliability validated using expression profiling*

*GTF2F2* was independently regulated by miR-708-5p, and the expression level of miR-708-5p was significantly different, so the expression level of *GTF2F2* should have a significant difference also. To confirm this, the expression of miR-708-5P and *GTF2F2* in paired lung tumor tissue samples and paraneoplastic tissue samples were downloaded from the TCGA database. The expression levels of miR-708-5p and *GTF2F2* were significantly higher in lung tumor tissue samples than in paraneoplastic tissue samples (Figs. 7A and 7B; Tables S5 and S6). Pearson correlation presented that there was a significantly positive correlation between miR-708-5p and *GTF2F2* in Fig. 7C ($r = 0.47$, $p = 1.63e{-}07$).

### *Evaluating the performance of the marker miRNA using external datasets*
Two external validation datasets were used to verify the performance of our marker miRNA. The expression level of miR-708-5p was significantly higher in NSCLC tissues than in normal tissues base on the GSE56036 and GSE36681 datasets (Figs. 8A and 8B; Tables S7 and S8). Additionally, the results of ROC analysis of the GSE56036 and GSE36681 datasets showed that miR-708-5p had good diagnostic values for NSCLC. The AUC of miR-708-5p was 0.835 in the GSE56036 dataset and 0.793 in the GSE36681 dataset (Figs. 8C and 8D).

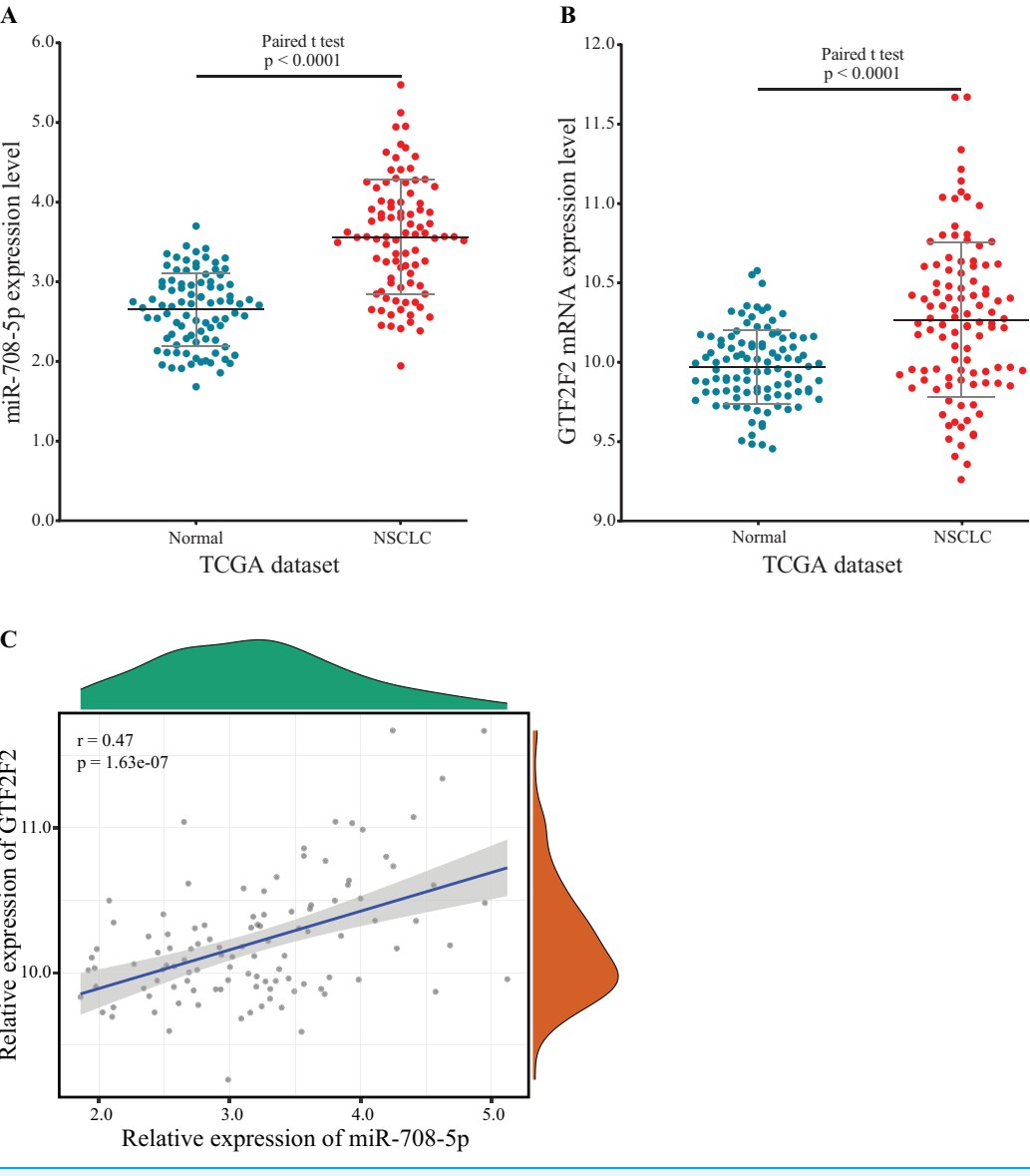

**Figure 7 The expression levels of *miR-708-5p* and *GTF2F2*.** (A) The expression of *miR-708-5p* in paired lung tumor tissue samples and paraneoplastic tissue samples from the TCGA. (B) The expression of *GTF2F2* in paired lung tumor tissue samples and paraneoplastic tissue samples from the TCGA. (C) The correlation between miR-708-5p and *GTF2F2* mRNA expression in the TCGA dataset.

### *Reliability validated by literatures*

By searching and synthesizing the related literatures of miR-708-5p, we found that miR-708-5p has been reported to be associated with lung cancer (Table 2).

Among the 10 retrieved papers related to lung cancer in Table 3, four papers did not state the relationship between miR-708-5p and lung cancer. One paper demonstrated their relationship with a bioinformatic approach, and in the rest five papers, the relationship between them was confirmed experimentally.

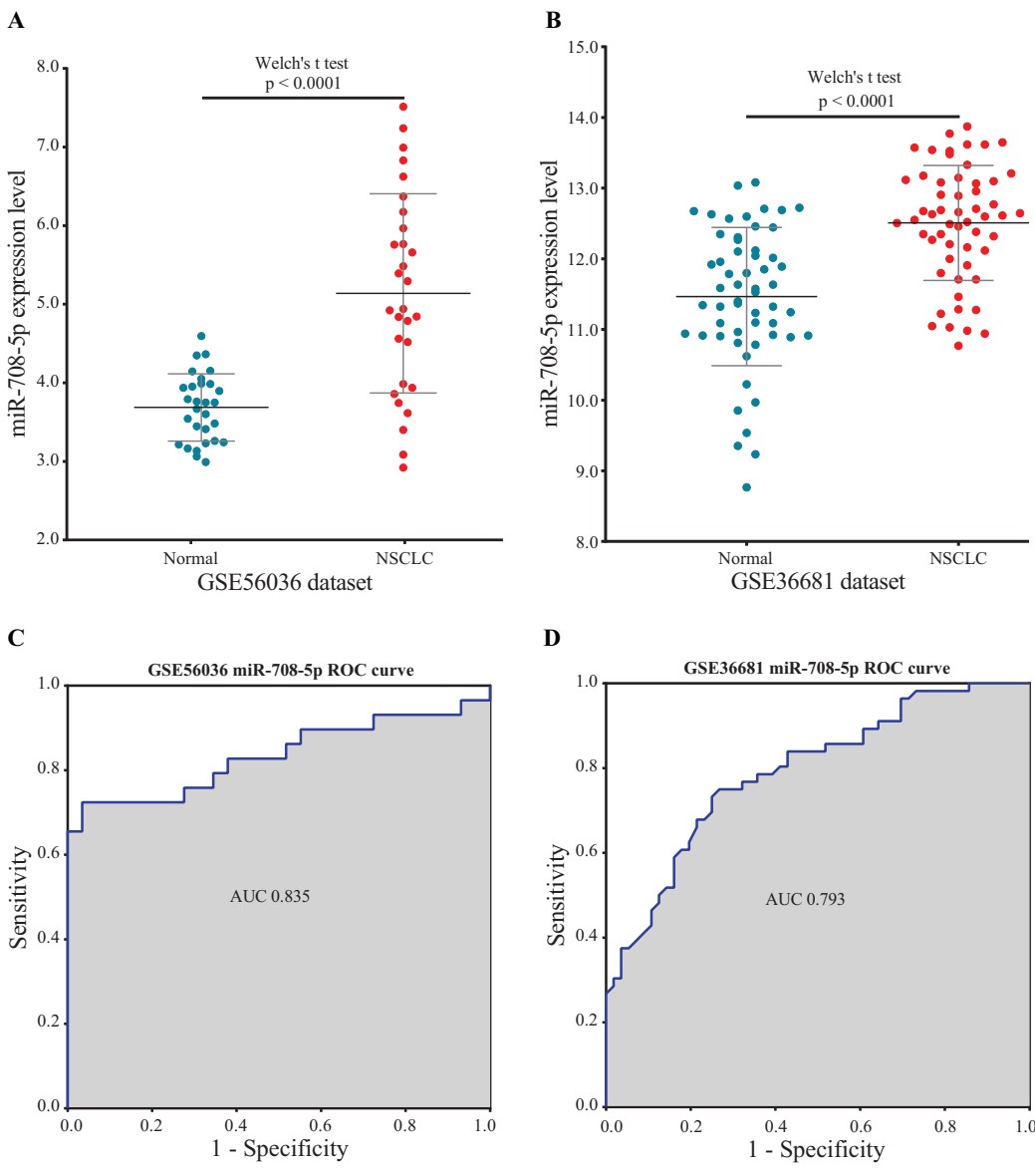

**Figure 8** *miR-708-5p* **expression levels and ROC curves in two external GEO datasets.** (A) The expression of *miR-708-5p* in the GSE56036 dataset. (B) The expression of *miR-708-5p* in the GSE36681 dataset. (C) ROC curve of *miR-708-5p* in the GSE56036 dataset. (D) ROC curve of *miR-708-5p* in the GSE36681 dataset.

Using immunoblotting, Dual-Luciferase reporter, and immunocytochemistry assays, *Liu et al. (2018)* found that miR-708-5p could be used as a new diagnostic and prognostic marker for NSCLC, the study showed that miR-708-5p directly inhibited the translation of *DNMT3A*, resulting in a significant decrease in genome-wide DNA methylation and upregulation of the tumor suppressor *CDH1*. The upregulation of *CDH1* reduced the activity of the Wnt/β-catenin signaling pathway, which in turn affected the stem cell properties of NSCLC cells, and in clinical analysis, patients with increased miR-708-5p expression had significantly higher survival and lower recurrence rates. In the study by *Xing et al. (2010)*, miRNA expression was detected by GeneChip miRNA arrays in lung

**Table 2 Results of the literature search of miR-708 in PubMed.**

| miRNA | Relevant literature | Cancer-related | Related to lung cancer |
|---|---|---|---|
| miR-708-5p | 108 | 73 | 10 |

**Table 3 Literatures about miR-708-5p associated with lung cancer in PubMed.**

| Year | Author | Journal | PMID |
|---|---|---|---|
| 2019 | Yang, Xia et al. | Int J Biol Sci | 31360113 |
| 2017 | Liu, Tianchi et al. | Clin Cancer Res | 28972040 |
| 2016 | Hu, Ling et al, | Oncotarget | 26870998 |
| 2015 | Wu, Xiaoping et al. | Oncotarget | 26678031 |
| 2014 | Molina-Pinelo, Sonia et al. | PLoS One | 24625834 |
| 2013 | Ryu, Seongho et al. | Cancer Cell | 23328481 |
| 2012 | Jang, Jin Sung et al. | Clin Cancer Res | 22573352 |
| 2012 | Pentheroudakis, George et al. | Clinical & Experimental Metastasis | 23124598 |
| 2010 | Xing, Lingxiao et al. | Mod Pathol | 20526284 |
| 2010 | Patnaik, Santosh K et al. | Cancer Res | 20028859 |

cancer tissue samples and RT-PCR in sputum, respectively, and all the results showed that miR-708-5p was highly expressed in lung cancer samples. *Jang et al. (2012)* used microarray data analysis, *RT-qPCR* and Luciferase reporter assay techniques to show that miR-708-5p was significantly more expressed in NSCLC tumor tissues than in paraneoplastic tissues. *Wu et al. (2016)* used qRT-PCR and Luciferase assay showed that miR-708-5p expression inhibited lung cancer invasion and metastasis *in vitro* and *in vivo*. To define distinct molecular features of these two major histological subtypes of NSCLC, *Molina-Pinelo et al. (2014)* detected miR-708-5p overexpression in squamous cell lung cancer compared to adenocarcinoma by TaqMan low-density array and qPCR methods.

These reports experimentally confirmed that miR-708-5p is highly expressed in lung cancer, containing NSCLC, can be used as a diagnostic and prognostic marker. Our finding is consistent with these experimental results.

## DISCUSSION

How does miR-708-5p play a role in lung carcinogenesis and development through independent regulation of the hub gene *GTF2F2*? A detailed analysis of its potential regulatory mechanism was explored here.

The general transcription factor complex GTF2 A, B, E, F and H were jointly involved in protein expression (*Flores, Ha & Reinberg, 1990*). Among them, *GTF2F* was a tetrameric molecule consisting of two subunits, *GTF2F1 (RAP74)* and *GTF2F2 (RAP30)* (*Thomas & Chiang, 2006*). Human *GTF2F* was widely expressed in various tissues and organs of the body, especially in the liver and lung. *GTF2F2* was involved in gene transcription initiation, promoter clearance, and elongation, and it was linked to *GTF2F1* via the N-terminus. The middle part of *GTF2F2* can bind to the 5th subunit of RNA polymerase II (Pol II), *RPB5*,

through RPB5-mediating protein (RMP), hence helping Pol II to bind to the promoter region and influence subsequent transcriptional elongation (*Jentsch et al., 2002*). At the C-terminus of *GTF2F2*, there was also a non-specific DNA-binding structural domain that possessed a characteristic ATP-dependent DNA unwinding enzyme activity. *GTF2F2* as a target molecule for many important transcription factors affected the formation or stability of the transcription initiation complex, thus influencing and regulating the transcription process.

It has been established that changes in *GTF2F2* function will lead to serious diseases (*Bansard et al., 2011*; *Kaplan & Stockwell, 2012*). However, the mechanism of *GTF2F* with cancer is not clear. The MYC protein family played an important role in normal physiology, proliferation and development, and its dysregulated expression in cancer was associated with poor prognosis and disease aggression (*Kalkat et al., 2017*). MYC functions as a master regulatory transcription factor that binds to and regulates the expression of thousands of target genes (*Cole & McMahon, 1999*). The MYC family encoded transcription factors containing six highly conserved regions, termed MYC homology boxes (MBs), among them, MB0 directly interacts with TFIIF in a transcription elongation complex, containing CDK9 and Pol II, MB0 is dispensable for tumor initiation but is a major accelerator of tumor growth (*Kalkat et al., 2018*). HBx was found to target RPB5 to stimulate transactivation (*Cheong et al., 1995*), RPB5-mediating protein (RMP) interacted with RPB5 and counteracts transactivation by HBx. TF2F2 has been shown to associate with pol II and recruit pol II to the promoter in the transcription initiation (*Garrett et al., 1992*; *Roeder, 1996*; *Sopta, Burton & Greenblatt, 1989*; *Wei et al., 2001*). TF2F1 binds to the initiation complex will allow pol II to make promoter contact (*Joliot, Demma & Prywes, 1995*; *Zhu, Joliot & Prywes, 1994*). The D5 region of RMP is necessary and sufficient for the association between RMP and GTF2F. Interaction with GTF2F is required for the suppression of activated transcription by RMP. RMP regulates the genes related to apoptosis and cell cycle, plays an antiapoptotic role in the proliferation and growth of HCC cells. *BCL2*, a gene that inhibits apoptosis, was elevated in overexpressing RMP (RMPo) tumors, while *BAX*, a gene that drives apoptosis, was decreased in RMPo tumors. Meanwhile, depletion of RMP induced G2 arrest in HCC cells by reducing the expression of Cdk1 and Cyclin B (*Yang et al., 2011*). From the above studies, we can conclude that TF2F can interact with different factors to influence the cell carcinogenesis process, for example, it binds to MB0 of MYC to promote cancer development, and binds to RMP to further inhibit the expression of pro-apoptotic genes and promote the expression of cell cycle-related genes to promote malignant tumorigenesis. It can be seen that if the expression of TF2F is regulated, it must be closely related to cell canceration. The schematic diagram of *GTF2F2* related functions is shown in Fig. 9.

It has also been demonstrated that miR-708-5p plays an important regulatory role in the development of lung cancer. *Jang et al. (2012)* reported that overexpression of miR-708-5p increased proliferation and invasion of lung cancer cells by experimental methods, and further identified a valuable link that miR-708-5p may directly downregulate *TMEM88* that weakened Wnt activity to promote lung cancer progression. In the contrary, the work of *Monteleone & Lutz (2020)* demonstrated that miR-708-5P exerted an inhibitory effect

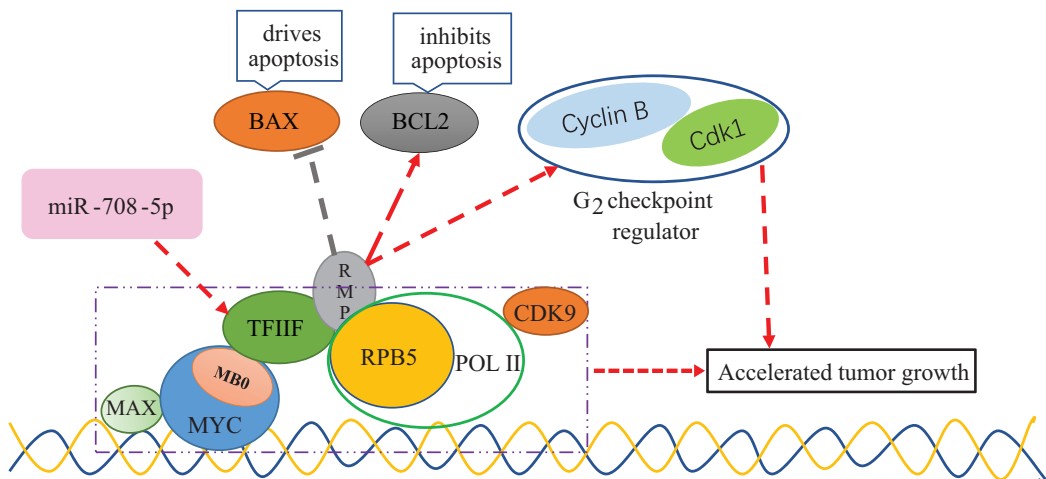

**Figure 9 The schematic diagram of *GTF2F2* related functions.** MB0 interacts directly with TFIIF, a general transcription factor that tightly associates with Pol II, enabling transcriptional elongation and accelerated tumor growth. TF2F2 binds to RMP to further inhibit the expression of pro-apoptotic gene (*BAX*) and promote the expression of cell cycle-related genes (*Cdk1* and *Cyclin B1*) to promote malignant tumorigenesis. Arrows indicate promotion and vertical bars indicate inhibition.

on lung carcinogenesis by suppressing Prostaglandin E2 (PGE2) signaling. Their work evidenced that miR-708-5p directly downregulated Cyclooxygenase-2 (COX-2) and microsomal prostaglandin E synthase 1 (mPGES-1). COX-2 and mPGES-1 promoted PGE2 synthesis in the arachidonic acid metabolism pathway.

Our results also showed that miR-708-5p had an important role in lung carcinogenesis and development, which was consistent with the results of existing studies, but the pathway of action was different from the existing studies. Our study found that in lung tissues, the gene of TF2F2 was independently regulated by miR-708-5p, and the gene of TF2F2 belonged to the core gene. As a core gene, the expression product of the TF2F2 gene was very important in the protein interaction network. At the same time, the regulation of this gene was very sensitive and fragile, and in lung tissue, it was only independently regulated by miR-708-5p. If the expression of miR-708-5p changes, the protein interaction network will be greatly affected, and further cellular functions will be greatly affected. In-depth analysis revealed that TF2F can interact with different factors to influence the cell carcinogenesis process, for example, it binds to MB0 of MYC to promote cancer development and binds to RMP to further inhibit the expression of pro-apoptotic genes and promote the expression of cell cycle-related genes to promote malignant tumorigenesis. The above mechanism has not been reported so far.

The limitation of this study should be considered. Since the strategy in this study was only used to identify non-small cell lung cancer miRNA markers and was not widely applied to the identification of other disease markers, the generalizability of the strategy needs to be further evaluated. Because network regulation in the human body is very complex, it needs to be evaluated with more extensive disease and data to see whether the

role of vulnerable nodes in biological networks can be influenced and weakened by other co-regulatory networks.

## CONCLUSIONS

This study proposed a novel idea for marker screening, that is, miRNAs can be used as biomarkers if they can independently regulate hub genes with important roles in the PPI network, and the expression products of the regulated hub genes play important roles in the signaling pathways of related tissue diseases.

In this study, for lung cancer, 15 SDE miRNAs were identified, and the SDE miRNAs with the top five AUC values and NOG ≥ 3 were selected as candidate markers, they were considered to have good sensitivity and specificity, and also have strong independent regulation abilities. Then, the importance of the target gene of candidate markers was investigated. The PPI network was constructed and biological function and pathway analysis were performed. The highly connected hub gene plays an important role in the biological processes, and generally, is an important target. A total of 61 hub genes regulated by these five candidate miRNAs were obtained. We found the important hub gene *GTF2F2*, which was independently regulated by miR-708-5p, and it interacted with RNA Pol II as a basal transcription factor to perform transcriptional functions and accelerate tumor growth. In a word, the gene *GTF2F2* has an important role in the induction and progression of lung cancer, at the same time, it is subject to a fragile independent regulation. Dysregulation of the fragile sole miRNA-gene interaction would inevitably lead to the occurrence and development of diseases. Therefore, miR-708-5p, the regulator of this gene, can be considered as a biomarker. Finally, the screening ability of miR-708-5p was evaluated by the AUC values of two external GEO datasets. The reliability of miR-708-5p was confirmed by expression level change of miR-708-5p and its target gene *GTF2F2* and by experimentally literature validation.

This study further explored the potential mechanism of miR-708-5p in lung cancer tissue. miR-708-5p plays a role in lung carcinogenesis and development through independent regulation of the hub gene *GTF2F2*. the disorder of the independent regulation is easy to cause dramatic changes in intracellular function and cause lung carcinogenesis and development. This mechanism has not been reported.

## ACKNOWLEDGEMENTS

Thanks to all the researchers and staff working for The Cancer Genome Atlas database and Gene Expression Omnibus database.

### Funding
The authors received no funding for this work.

### Competing Interests
The authors declare that they have no competing interests.

## Author Contributions

- Renqing Nie conceived and designed the experiments, performed the experiments, prepared figures and/or tables, authored or reviewed drafts of the paper, and approved the final draft.
- Wenling Niu performed the experiments, authored or reviewed drafts of the paper, and approved the final draft.
- Tang Tang analyzed the data, authored or reviewed drafts of the paper, and approved the final draft.
- Jin Zhang analyzed the data, authored or reviewed drafts of the paper, and approved the final draft.
- Xiaoyi Zhang conceived and designed the experiments, prepared figures and/or tables, authored or reviewed drafts of the paper, and approved the final draft.

## Data Availability

Data is available at NCBI GEO: GSE102286, GSE56036, GSE36681 and TCGA dataset (TCGA-LUAD, TCGA-LUSC,The Cancer Genome Altas, https://www.cancer.gov).

The raw measurements are available in the Supplemental Files.

## Supplemental Information

Supplemental information for this article can be found online at http://dx.doi.org/10.7717/peerj.12369#supplemental-information.

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
