# Peer review of "Integrating microRNA expression, miRNA-mRNA regulation network and signal pathway: a novel strategy for lung cancer biomarker discovery"

_PeerJ, doi:10.7717/peerj.12369_

## Round 0.1 · original submission · Major Revisions

Dear Dr. Nie,

The manuscript has been reviewed by four reviewers and they feel that this needs to be revised considerably. All the reviewers feel that there are some changes that could make the manuscript strong. Especially, Reviewer 2 has suggested a correlation plot between miR-708-5p and GTFF2 using the TCGA database to support your results. Please revise the manuscript accordingly.

Reviewer 1 ·

Basic reporting

no comment

Experimental design

no comment

Validity of the findings

no comment

Additional comments

This is a manuscript by Nie et al that elucidates a new strategy for lung cancer marker screening by integrating microRNA expression, regulation networks and signal pathways. The authors find miR-708-5p as a biomarker. This manuscript is a step forward in the framing of policies for lung cancer diagnosis. It is written in a very simple and easy to follow manner. The conclusions are supported by the analysis. This manuscript can be accepted as is at PeerJ- a few minor comments

1) Please include study limitations in the discussion
2) The figure legends should be expanded
3) Please check for grammar

Reviewer 2 ·

Basic reporting

The authors have constructed a human miRNA-mRNA regulatory network and have found that miR-708-5p independently regulates the hub gene GTF2F2 in non-small cell lung cancer. Gene Ontology analysis and KEGG pathway analysis were used to identify genes involved in cancer-related pathways.
TCGA database, GEO database, volcano plot and signal-to-noise ratio methods were used to obtain significant differentially expressed (SDE) miRNAs. Interestingly, they have found that GTF2F2 binds to polymerase II, which in turn regulates the transcription promoting tumor growth.
The authors have claimed that miR-708-5p could be used as a potential biomarker and its oncogenic role has already been proved experimentally in the previous literature studies. The mechanism of action reported by them is quite novel

Experimental design

No comment

Validity of the findings

No comment

Additional comments

Although, the mRNA expression of miR-708-5p and GTF2F2 are upregulated in NSCLC, can authors show a correlation plot between miR-708-5p and GTFF2 using the TCGA database.

·

Basic reporting

The language of the manuscript is clear and concise. Images presented are good quality and the manuscript is well-structured.

Experimental design

The hypothesis laid down by the authors for the identification of miRNA to be used a lung cancer marker based on downstream targets and pathways is reasonable. The datasets identified are rigorous and methods have been described well.

Validity of the findings

The conclusion derived from the findings does not take into account all the findings.

Additional comments

A documents with figure legends should have been provided with the submission.

Reviewer 4 ·

Basic reporting

No comment

Experimental design

no comment

Validity of the findings

no comment

Additional comments

Just as a minor comment, it would have been nice if the authors could add a schematic diagram of the mechanism through which miR-708-5p plays role in lung carcinogenesis.

---

## Round 0.2 · accepted · Accept

Dear Dr. Nie,

Your manuscript is now deemed acceptable by peerJ

Thanks

Reviewer 2 ·

Basic reporting

No comment

Experimental design

No comment

Validity of the findings

No comment

Additional comments

The authors have incorporated the suggestions, hence the manuscript can now be accepted.